# Impacts of Underground Reservoir Site Selection and Water Storage on the Groundwater Flow System in a Mining Area—A Case Study of Daliuta Mine

**Yanmei Chen** [1,2], **Yu Zhang** [1], **Fang Xia** [3], **Zhao Xing** [1] **and Licheng Wang** [1,*]

1   College of Resources and Environment, Yangtze University, Wuhan 430100, China
2   State Key Laboratory of Water Resource Protection and Utilization in Coal Mining, Beijing 102209, China
3   Wuhan Gathering Wisdom Environmental Protection Technology Co., Ltd., Wuhan 430079, China
*   Correspondence: 202071468@yangtzeu.edu.cn

**Abstract:** The natural ecological conditions in the Shendong mining area in China are very fragile, and water resources are seriously lacking. As the production scale of the Shendong mine continues to expand, the demand for water for production and living is also increasing; however, the available surface and underground water is decreasing, and the water scarcity in the Shendong mine is becoming increasingly apparent. Obtaining water resources is a major technological challenge for green mining. The underground reservoir is a new type of underground water conservancy project, and the water shortage in China's coal mine pits is resolved by underground reservoirs, which also makes a substantial contribution to the effective utilization of water resources. How the construction of underground reservoirs affects the groundwater system in a mining area has become one of the most important factors to consider when finding sites for underground reservoirs. In this study, we took the Daliuta Coal Mine as an example. A numerical model based on the hydrogeological conditions in the mining area was developed to determine the effects on groundwater using FEFLOW software via the finite element method. The model was used to analyze the impacts of coal seam mining thickness, overlying lithology, water-storage range and the water level of the underground reservoir on the groundwater flow system in the mining area. The results indicate that the thickness of the coal seam mining and the lithology of the overlying reservoir both had a significant effect on the upper aquifer system. The water-storage range and water level of the underground reservoirs were the main influences on the lower aquifer system. The results prove that underground reservoir storage had a good effect on water retention in the groundwater system in the mining area, and was able to achieve the desired result of storing groundwater and reducing water loss. It also had a positive feedback effect on the mining area's environment.

**Keywords:** coal mine underground reservoir; groundwater flow system; determinants; numerical simulation

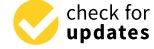



## 1. Introduction

The construction of China's large-scale energy infrastructure and the shift in coal mining to western China has put further strain on the already fragile ecological environment and limited water resources. The shortage of water resources in western China has become a key factor restricting the scientific development of the coal industry [1]. After the mining of close coal seam groups, the overlying lithology will be deformed and destroyed under the actions of the weight load and overlying rocks, forming the "upper three zones" (caving zone, water-flowing fractured zone, bending zone) [2]. Based on their studies of the variation law of groundwater systems before, during, and after coal mining in western mining areas, a team led by Gu proposed the concept of using a coal mine's underground reservoir to store and utilize water resources in the mining area [3,4]. The underground reservoir of a coal mine consists of a goaf, a caving zone, a barrier pillar, an

artificial dam, etc. [5,6]. The construction of an underground reservoir not only results in the protection and recycling of water resources, but also alters the original aquifer structure in the mining area and the conditions for the recharge and discharge of groundwater. It affects the water quantity and quality of the regional groundwater system and its relationship with the environment. In recent years, as the technology of electronic computers has improved, numerical simulations have been widely used to study groundwater-related problems in mining areas. Rudakov et al. [7] presented and validated an analytical model of water inflow and rise in a flooded mine, and a realistic prediction of transient mine water rebound and inflow into a mine with layers of heterogeneous rocks. Rudakov et al. [8] developed criteria and a calculation methodology with which to realistically evaluate the parameters of the efficiency of operation for open non-return and circulation geothermal systems. Sadovenko et al. [9] established a hydrochemical model of a region for the evaluation of groundwater contamination. Bazaluk et al. [10] established a hydrodynamic model and made it possible to improve the reliability of hydrodynamic prognoses and develop technological schemes to control water after a mine closure.

Numerous studies on the theory of the development of the "upper three zones" and the laws of water storage and seepage in underground reservoirs have been carried out. Pang et al. [11] used the Universal Distinct Element Code numerical calculation model to calculate the height of the water-flowing fractured zone. Li et al. [12] analyzed the mechanism for the evolution of overlying fissures and proposed the "fissure similarity" method to evaluate the boundary range of overlying fissures. Shi et al. [13] used simulation experiments to study the fracture, the movement, and the law of fracture development in the overlying rock structure in a coal mining area. Through the solid–liquid coupling similarity model test, Li et al. [14] obtained the law of the development of overlying fissures and the characteristics of groundwater seepage in a mining area. Ma et al. [15] used a numerical simulation to study the groundwater seepage characteristics in the broken rock mass in a goaf. Guo et al. [16] investigated the impacts of geological and mining factors on the movement and damage of overlying rock and the height of a water-flowing fractured zone using a numerical simulation method. Additionally, Gao et al. [17] summarized the relationship between the water storage coefficient and height under typical mining conditions through numerical analysis. With the expansion of research, how the location of the underground reservoir and the water storage impact the groundwater flow system in a mining area have become key considerations in respect of underground reservoir construction.

Site selection and water storage factors, including the coal seam mining thickness, the overlying lithology, the water-storage range, the water storage level, etc., have significant impacts on the groundwater system in a mining area. These are some of the key issues that need to be urgently addressed in the siting of groundwater reservoirs and the protection of regional groundwater environmental systems. In this study, using the Daliuta Coal Mine as the research object, we used FEFLOW software to construct a groundwater flow numerical model of the mining area, including the underground reservoir, and then discussed the impacts of various site selection and water storage factors on the groundwater system. The results are expected to provide a reference for coal science development by assisting with the site selection, storage capacity design, and scheduling of underground reservoirs in coal mining areas of the same type.

## 2. Case Study

### 2.1. Hydrogeological Conditions of the Study Area

Daliuta Coal Mine is located in Daliuta Town, Shenmu County, Yulin City, Shaanxi Province, China, which is one of 30 mining areas in the Shendong Mining Area, in the transition zone between the northern Loess Plateau and the southeastern margin of the Mu Us Desert. Figure 1 shows the geographic location of the site under study. The terrain is generally elevated in the west, low in the east, elevated in the north, and low in the south. The elevation is generally 1000–1250 m. The maximum is 1358 m, and the minimum is

1085 m; there is little variation at the ground level. The mining area is dominated by a large, wide, and gentle syncline structure, with no developed fault folds. There is only one fault in the north of the study area, without magmatic activity, and it belongs to a simple structural area. The only fault in the study area is called the F6 fault. The study area has an arid continental monsoon climate with frequent droughts, little rain, and strong evaporation; rainfall is concentrated in July and August. On the west side of the study area is the Wulan Mulun River, and on the east side is the Boniu River. The Wulan Mulun River flows from the northwest to the southeast, through the west side of the mining area, and the Boniu River flows from north to south through the southeastern edge of the mining area. Other tributaries, such as the Muhe stream, the Wangqu stream, and the Shuang stream are also present.

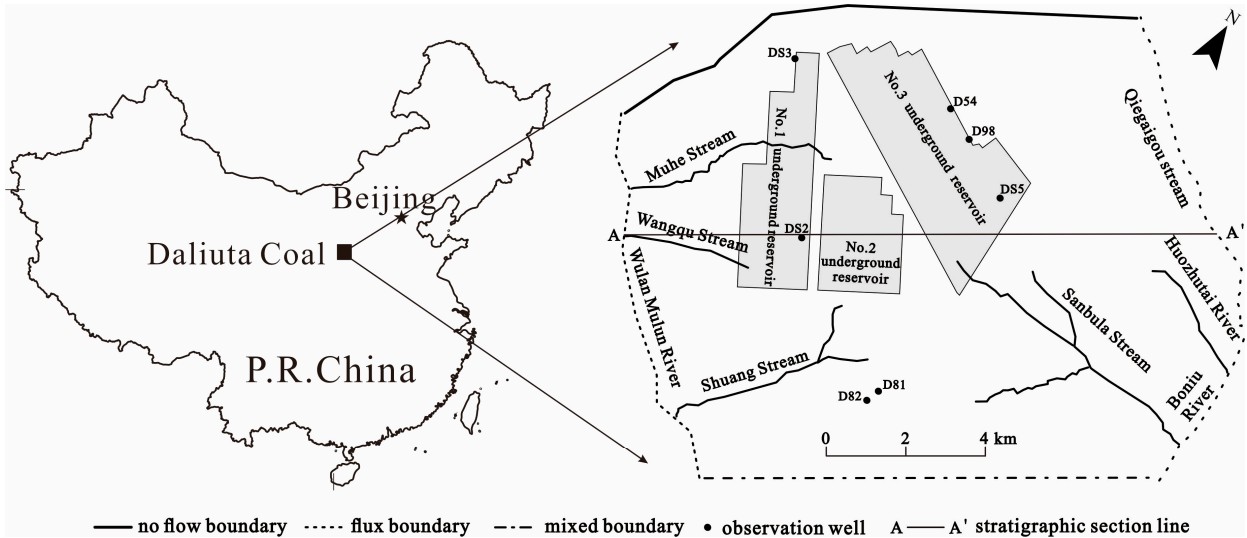

**Figure 1.** Schematic diagram of model boundary.

The strata in the study area are classified as the Yanhe strata area within the Ordos division of the North China strata. The Mesozoic strata are mainly exposed by Jurassic strata, and Quaternary strata are widespread. In the study area, the aquifer and aquitard are distributed alternately. According to the lithologic combination, pumping test, and water-bearing characteristics of the strata, they can be divided into the coal seam, aquifer, and aquitard (Table 1). Figure 2 shows the stratigraphic section of the site under study.

The groundwater aquifer system is a water-bearing rock series with uniform hydraulic connections, which are trapped by water-resisting or somewhat-water-resisting strata. According to the lithologic characteristics of strata and the occurrence of groundwater in aquifer media, the groundwater in the mining area can be divided into two aquifer systems: the pore aquifer in Cenozoic loose rocks (Model layers 1–3, hereinafter referred to as the upper aquifer system) and the fissure aquifer in Mesozoic sedimentary rocks (Model layers 4–13, hereinafter referred to as the lower aquifer system). Under natural conditions, atmospheric precipitation recharges the upper aquifer system, then most of the groundwater discharges to the river, and a small part flows through the lower aquifer system via vertical leakage.

In summary, the study area is characterized as a heterogeneous, horizontally isotropic, vertically variable, spatially three-dimensionally structured, unsteady flow groundwater flow system.

**Table 1.** Model hierarchy.

| Model Layer Number | Stratigraphic Unit | Lithological Features | Average Thickness (m) | Groundwater Occurrence Type | | Hydraulic Conductivity (m/d) |
|---|---|---|---|---|---|---|
| 1 | Holocene Series, Upper Pleistocene Series | Aeolian sand, alluvial sand and gravel, sandy soil | 35 | Pore-phreatic water (water-rich medium) | | 35 |
| 2 | Middle Pleistocene Series | Loess, sub-clay, gravel layer | 15 | Aquifuge | Upper aquifer system | 0.1 |
| 3 | Lower Pleistocene Series | Gravel, sandy clay, coarse sand | 18 | Pore-phreatic water (Water-rich medium) | | 15.6 |
| 4 | Zhiluo Group | Siltstone, fine-grained sandstone | 14.8 | Fissure-phreatic water–confined water (water-rich weak) | | 0.2 |
| 5 | | Mudstone | 10 | Aquitard | | 0.025 |
| 6 | | 1–2 coal | 3 | Mineable coal seam | | 0.1 |
| 7 | | Mudstone | 7.8 | Aquitard | Lower aquifer system | 0.025 |
| 8 | | Fine sand-siltstone | 13.2 | Aquifer (water-rich weak) | | 0.2 |
| 9 | Yanan Group | Mudstone | 4.4 | Aquitard | | 0.025 |
| 10 | | Fine sand–siltstone | 10.2 | Aquifer (water-rich weak) | | 0.2 |
| 11 | | Mudstone | 3.2 | Aquitard | | 0.025 |
| 12 | | 2–2 coal | 4.2 | Mineable coal seam | | 0.1 |
| 13 | | Mudstone | 4.6 | Aquitard | | 0.025 |

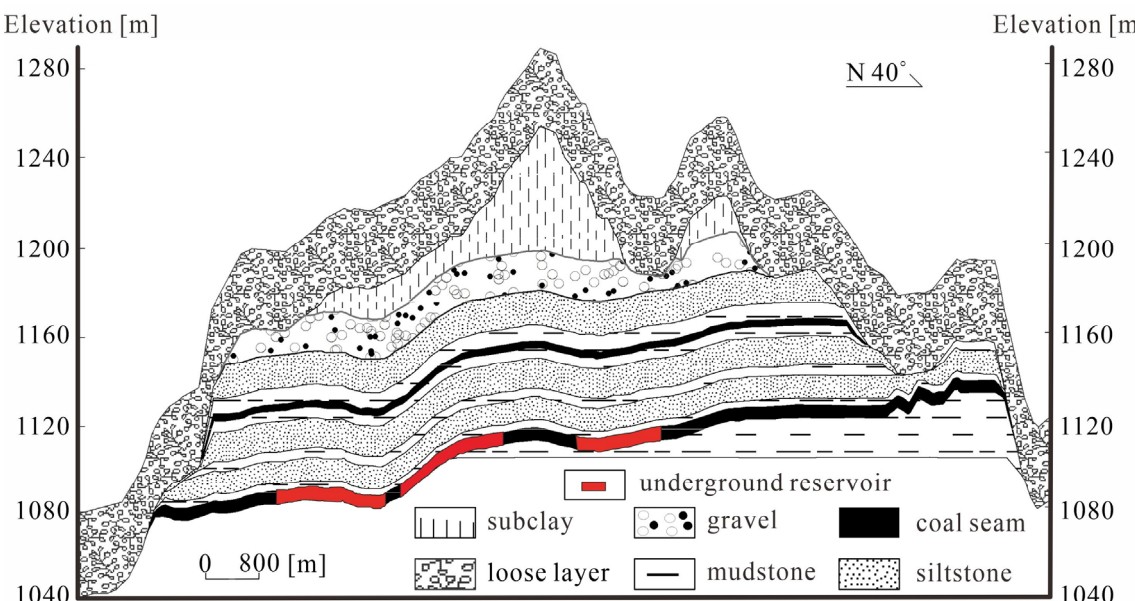

**Figure 2.** AA' stratigraphic section map.

## 2.2. Impact of Underground Reservoir Construction on the Hydrogeological Conditions of the Mining Area

The 2–2 coal seam in the study area is mainly used for long-arm mining, with a span of 240 m and a driving length of several kilometers. During this mining period, 20–30 m coal pillars are left. With the continuous mining of the coal seam, the goaf continues to increase, and the coal seam gradually becomes an aquifer. The "upper three zones" produced by underground coal mining are the main locations for the storage and transport of groundwater. Through the water-flowing fractured zone, the upper surface water and groundwater infiltration into the underground reservoir is allowed by the goaf and the caving zone. The original water-resistant coal seam has become an aquifer due to the creation of coal mines and the construction of underground reservoirs, and a groundwater depression cone centered on the underground reservoir has formed. Groundwater runoff changed from horizontal to vertical. In 2010, the underground reservoir in the 2–2 coal seam in the study area was put into service. Atmospheric rainfall infiltration is still the primary means of recharging the groundwater system, which discharges towards the underground reservoir.

### 3. Construction of the Groundwater Numerical Model in the Study Area

*3.1. Construction of a Groundwater Numerical Model in the Mining Area*

　　The simulation model of the Daliuta Coal Mine ranges 10.5–13.9 km from east to west and 9.1–10.5 km from north to south, making a total area of 126.8 km² (Figure 1). There are 18 mines in the Shendong mining area. Groundwater is mainly discharged through the pit. The watershed formed between the mines divides each mine into a separate hydrogeological unit. The study area is equipped with four upper-aquifer-system water-level monitoring wells (D81, D82, D54, D98) and three lower-aquifer-system water-level-monitoring wells (DS2, DS3, DS5). The eastern and western boundaries are flux boundaries, comprising the Boniu River and Wulan Mulun River, respectively. The southern boundary is a mixed boundary formed by the watershed of Huangtuliang Hill, and so is the northern boundary. The F6 fault, which is filled with sandstone and siltstone, forms the northern no-flow boundary. Vertically, the top of the simulation area is the recharge boundary of rainfall infiltration, and the bottom layer is characterized as the no-flow boundary.

　　The model was divided into 13 layers in the vertical direction based on the stratigraphic lithologic properties (Figure 3 and Table 1), and according to the research of Gao [17] and the pumping test in the mining area, the initial permeability coefficient values of each layer are shown in Table 1. According to the original data from the mining area and the topographic features, and referring to Gu's [18] research results, the infiltration coefficients of precipitation for the study area were divided and allocated. Figure 4 illustrates the division of the infiltration coefficients of precipitation.

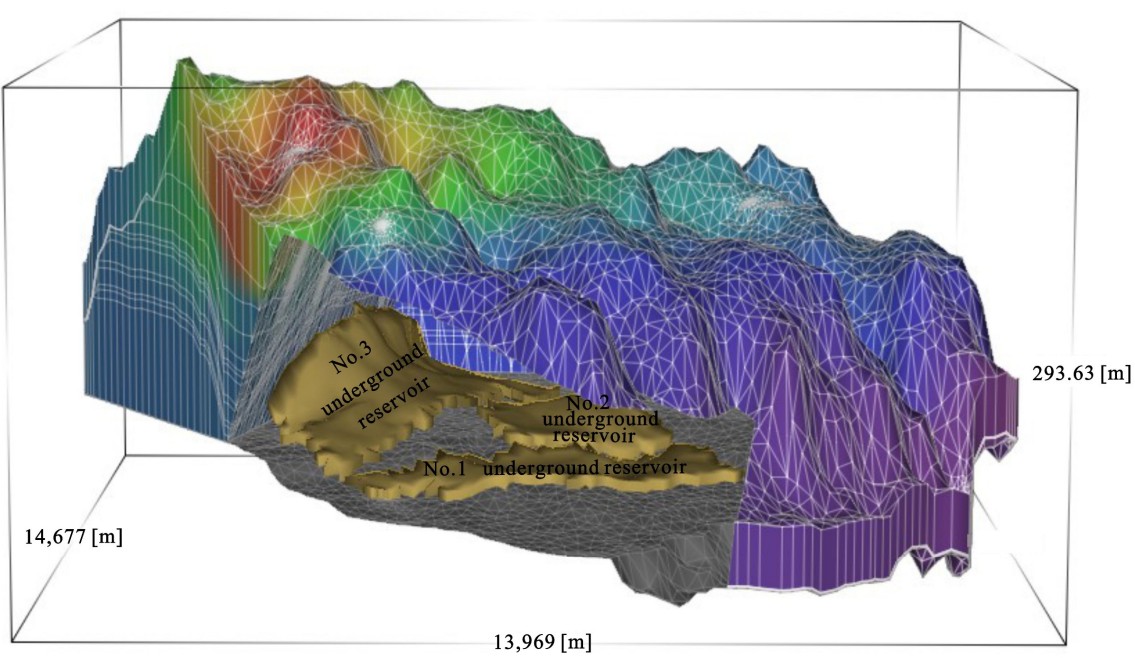

**Figure 3.** Simulation model of the groundwater system in the study area.

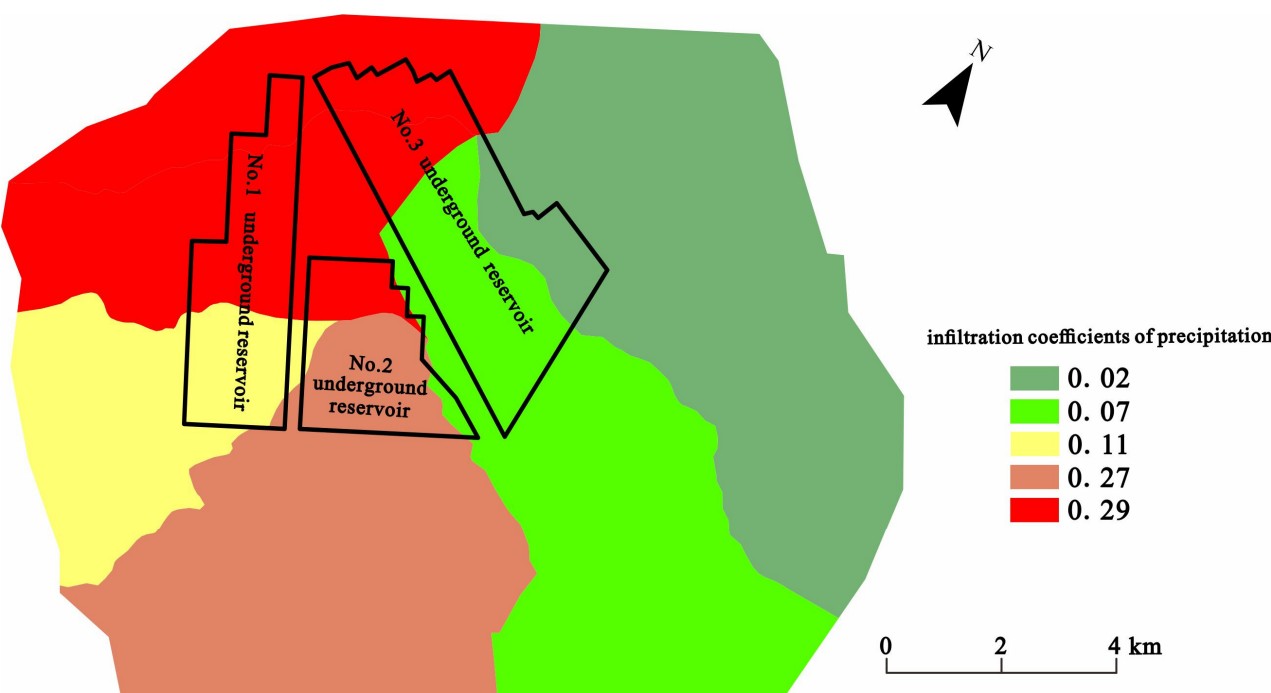

**Figure 4.** Division of infiltration coefficients of precipitation in the study area.

*3.2. Construction of Numerical Underground Reservoir Model*

3.2.1. Generalization of the Underground Reservoir Numerical Model

The water storage in the goaf or caving zone is characterized as a saturated medium with a constant water head, which is a fixed hydraulic head boundary, and the water-head value is determined by the water storage level. The goaf without water storage is characterized as an unsaturated medium. Above the goaf, the aquitard from the water-flowing fractured zone moved into a permeable layer, which resulted in an increase in water flowing from the upper aquifer system into the goaf and the water level of the upper aquifer system dropping until it reached the level of the water-resisting floor of the aquifer [1]. Taking the water-conducting fracture zone as the equivalent fractured medium, it is considered that the flow of groundwater in it conforms to Darcy's law. The contact zone between the water-flowing fractured zone and the aquifer is defined as the seepage boundary, and the boundary constraint condition is that the minimum groundwater head is higher than the bottom elevation of the aquifer; that is, when the aquifer water-level drops to the bottom of the aquifer, the groundwater in the aquifer will no longer flow to the underground reservoir [17,19].

The thickness of the 2–2 coal seam is between 0.97 and 7.43 m, the average thickness is 4.20 m, and the upper lithology is medium to hard. The calculation of the caving zone and water-flowing fractured zone according to the equivalent empirical formula [13], and the results are illustrated in Table 2. These results are comparable to the physical similarity model test outcome for the research area obtained by Shi et al. [13], and it is expected that the empirical formula approach can be utilized to determine the development of the "upper three zones" under varying coal seam mining thicknesses and lithologies.

**Table 2.** The heights of the caving zone and the water-flowing fissure zone were determined by empirical calculations and physically equivalent models.

| Name | Empirical Formulas | Overlying Lithology | The Formula Evaluates the Result (m) | Physically Similar Model Test Results (m) | | |
|---|---|---|---|---|---|---|
| | | | $\dfrac{\text{Minimum Value} \sim \text{Maximum Value}}{\text{Average Value}}$ | No. 1 Reservoir | No. 2 Reservoir | No. 3 Reservoir |
| Caving zone | $H_m = \dfrac{100\sum M}{2.1\sum M+16} \pm 2.5$ | hard | | | | |
| | $H_m = \dfrac{100\sum M}{4.7\sum M+19} \pm 2.2$ | medium-hard | $\dfrac{6.28 \sim 15.98}{12.91}$ | 12~14 | 10~12 | 9~12 |
| | $H_m = \dfrac{100\sum M}{7.0\sum M+63} \pm 1.2$ | very weak | | | | |
| Water-flowing fractured zone | $H_{li} = \dfrac{100\sum M}{1.2\sum M+2.0} \pm 8.9$ | hard | | | | |
| | $H_{li} = \dfrac{100\sum M}{1.6\sum M+3.6} \pm 5.6$ | medium-hard | $\dfrac{24.29 \sim 53.57}{45.81}$ | 55~60 | 33~35 | 33~35 |
| | $H_{li} = \dfrac{100\sum M}{5.0\sum M+8.0} \pm 3.0$ | very weak | | | | |

Note: $\sum M$ is the cumulative mining thickness; the number after $\pm$ is the medium error; $H_m$ is the height of the caving zone; $H_{li}$ is the height of the water-flowing fractured zone.

Combined with the research results of Shi et al. [13], Gao et al. [17], Li et al. [20,21], Qian et al. [22], the hydrogeological parameters of upper underground reservoirs in the 2–2 coal seam were calculated, as shown in Table 3. The reservoir's water storage coefficient varies along with its water level. Referring to the previous research results for the mining area, we determined that the value of the water storage coefficients of the three underground reservoirs in the study area ranges from 0.25 to 0.35, and there is a relationship between the water storage coefficient and the reservoir's water level, which decreases with the increase in height [5,22–24].

**Table 3.** The hydraulic conductivity of the actual groundwater reservoir and the upper water-flowing fractured zone in the mining area.

| Stratigraphic Number | | No. 1 Underground Reservoir | | No. 2 and No. 3 Underground Reservoirs | |
|---|---|---|---|---|---|
| | | Kzz (m/d) | Specific Storativity | Kzz (m/d) | Specific Storativity |
| 4 | | 53.568 | 0.0010 | | |
| 5 | | 54.432 | 0.0010 | 53.56800 | 0.0010 |
| 6 | water-flowing fractured zone | 58.16 | 0.0010 | 56.1600 | 0.0010 |
| 7 | | 95.04 | 0.0010 | 95.0400 | 0.0010 |
| 8 | | 190.080 | 0.0010 | 190.0800 | 0.0010 |
| 9 | | 198.720 | 0.0035 | 198.7200 | 0.0035 |
| 10 | caving zone | 368.064 | 0.2074 | 368.0640 | 0.2074 |
| 11 | | 397.440 | 0.2226 | 379.2960 | 0.2226 |
| 12 | goaf | 570.240 | 0.2826 | 570.2400 | 0.2826 |

Note: Kzz is the hydraulic conductivity coefficient in the z direction.

### 3.2.2. Model Identification and Verification

The initial conditions need to be given for a numerical model in order to solve the unsteady groundwater flow problem. In the simulation process, the steady flow field under natural conditions is calculated as the initial flow field. The simulation results are consistent with Gu [23], indicating that the model generalization and parameter settings are reasonable (Figure 5).

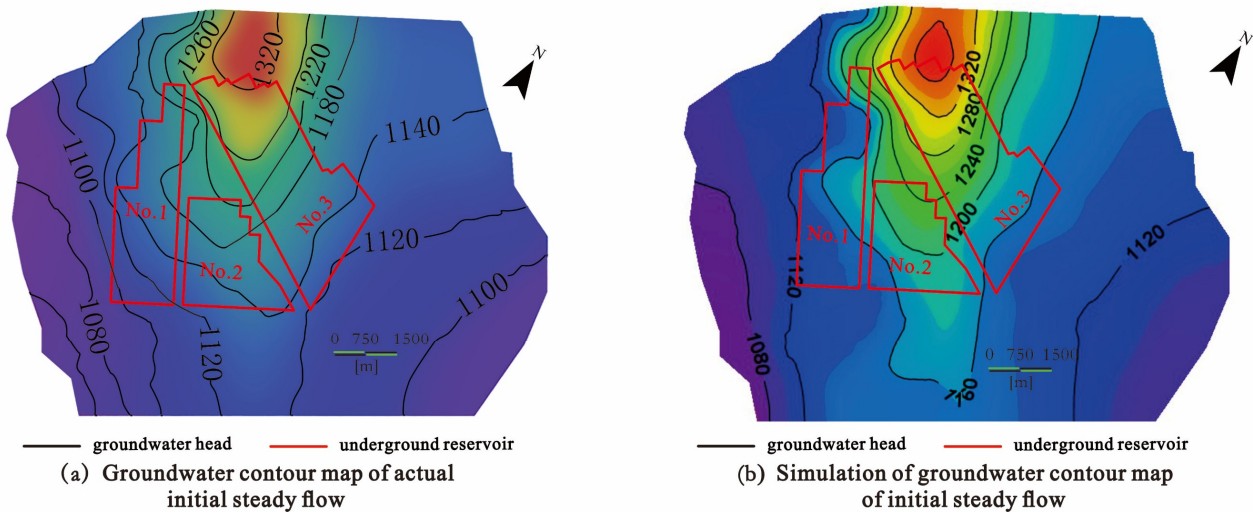

**Figure 5.** Comparison of actual and simulated initial steady-flow groundwater-level isolines.

In order to verify the rationality of the parameter settings after the completion of the groundwater reservoir, the measured groundwater-level dynamic curves of four monitoring wells (D81, D82, D54, and D98) were selected for fitting. The fitting period was from 1 January 2011, to 31 December 2014. The results are displayed in Figure 6. The figure illustrates that there were relative errors of 0.27%, 0.13%, 0.23%, and 2.65% between the water level observed in each hole, respectively, and the simulated water level during the identification period. It is considered that the parameter settings in the identification period of the model are reasonable.

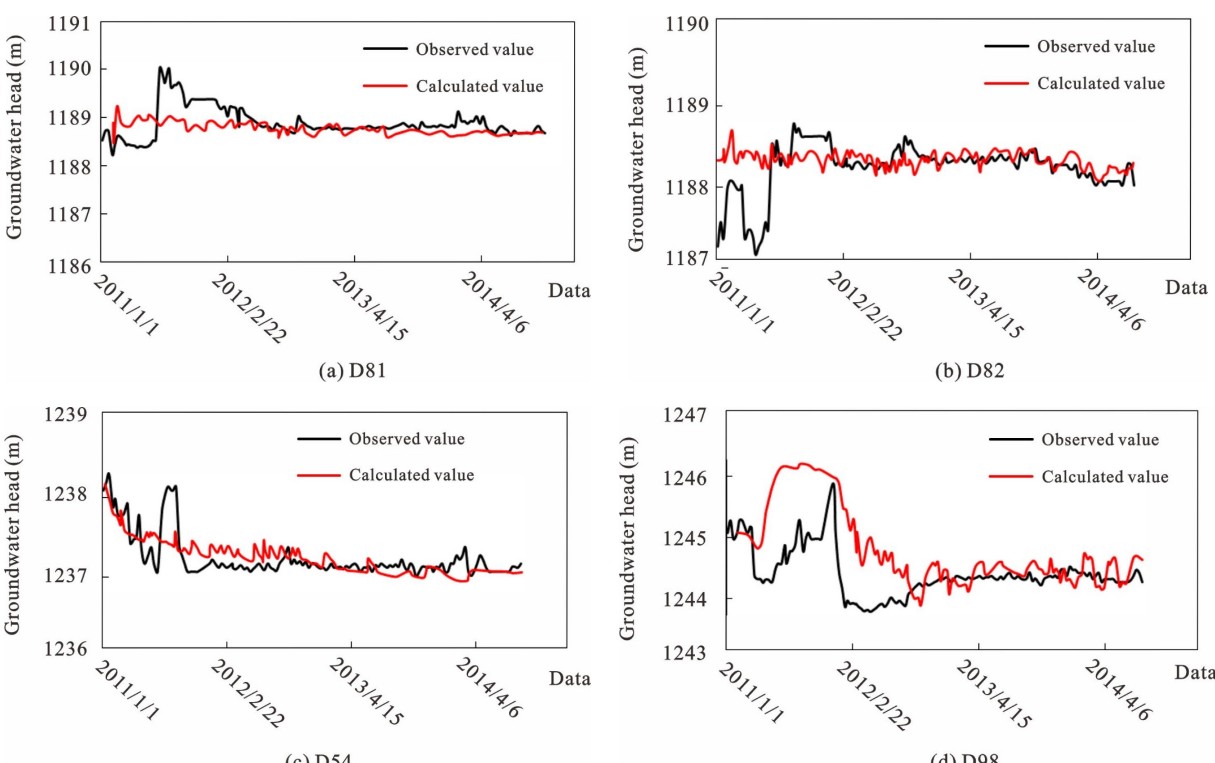

**Figure 6.** Comparison curves between observed and calculated groundwater levels: (**a**) D81, (**b**) D82, (**c**) D54, (**d**) D98.

In order to further test the accuracy of the model, the period from 1 January 2015 to 31 December 2018 was selected as the validation period, and the measured groundwater-

level dynamics of the DS2, DS3, and DS5 monitoring holes were compared with the simulated water level. Figure 7 illustrates that for the majority of the observed holes, accounting for more than 85% of the known water-level nodes, the absolute error between the calculated head and the actual head at each time point is within 2 m. This shows that the numerical model of the mining underground reservoir can be used to study the impacts of the site selection and storage factors of the underground reservoir on the groundwater flow system.

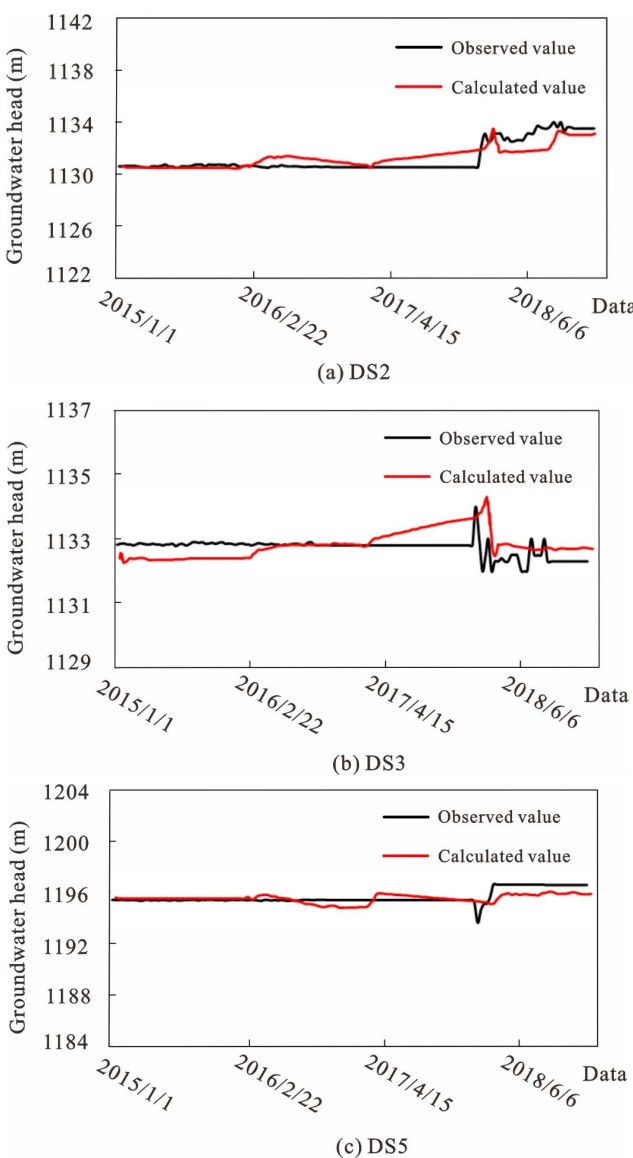

**Figure 7.** Comparison curves between observed and calculated groundwater levels: (**a**) DS2, (**b**) DS3, (**c**) DS5.

## 4. Impact of Different Site Selection Factors and Storage Factors on the Groundwater Flow System

The coal seam mining thickness, the overlying lithology, the water-storage range, and the water level of the underground reservoir are the factors that have the most influence on the groundwater flow system in the mining area [18,23]. In order to identify the impacts of these factors on the groundwater flow system, the mining-area underground reservoir model was used to simulate and analyze the factors.

The coal seam mining thickness and the lithology above it impact the development of the "upper three zones". The development height of the "upper three zones" was

determined using the empirical formula in Table 2; by changing the hydraulic conductivity and storage coefficient in the range of the "upper three zones", the different mining thicknesses and lithology of the coal seam were realized. We changed the number of boundaries of fixed-water-level nodes of the 2–2 coal seam in the model to represent the various reservoir water-storage ranges. In respect of the reservoir water storage height being different, the change in water storage level was simulated by changing the water storage coefficient of the underground reservoir.

### 4.1. Impact of Coal Seam Mining Thickness

According to the 2–2 coal seam mining thickness and other coal seams in the study area, the maximum mining thickness is 1 m and the minimum mining thickness is 4.2 m. Therefore, we selected mining thicknesses of 1, 3, and 4.2 m for comparative study, and the simulation results are shown in Figure 8.

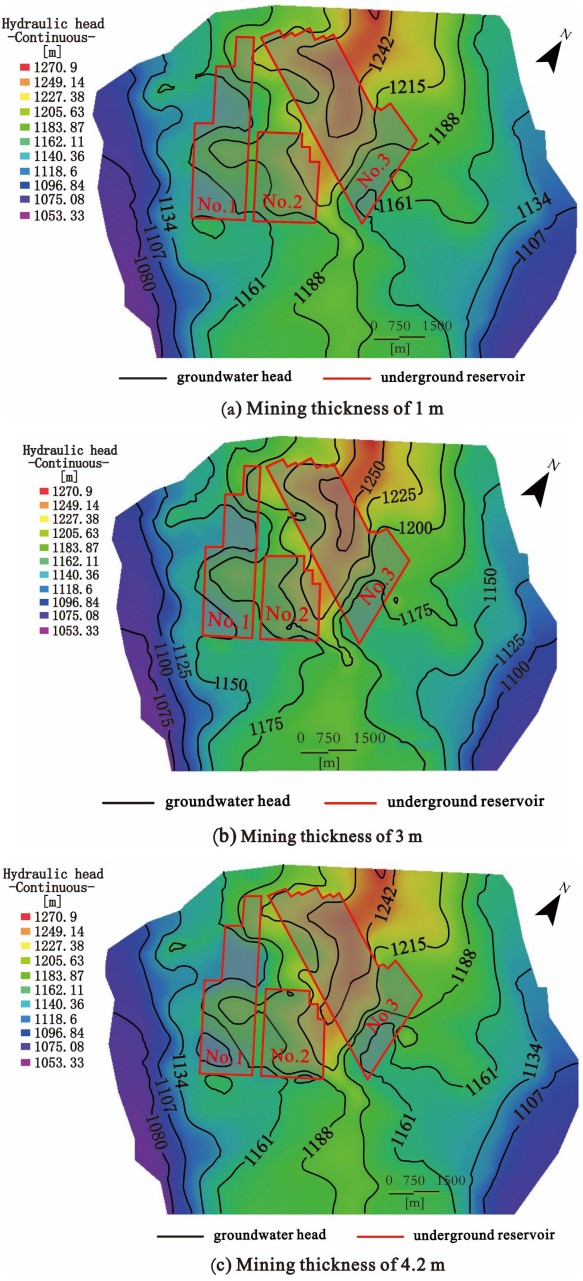

**Figure 8.** Groundwater contour map of the upper aquifer system under different thicknesses of coal seam: (**a**) 1 m, (**b**) 3 m, (**c**) 4.2 m.

It can be seen in Figure 9 that with an increase in the coal seam mining thickness, the groundwater level of the upper aquifer system shows a downward trend under the condition of goaf water storage. As the thickness of the coal seam mining increases along with the development height of the "upper three zones", the conditions of the underground reservoir's water flow channel improve, and the source of water inrush increases. Combined with the water-level fluctuations in D54 and D98 (Figure 8), aquifer drainage occurs in some regions when the seam mining thickness of the upper aquifer system is greater than 3 m.

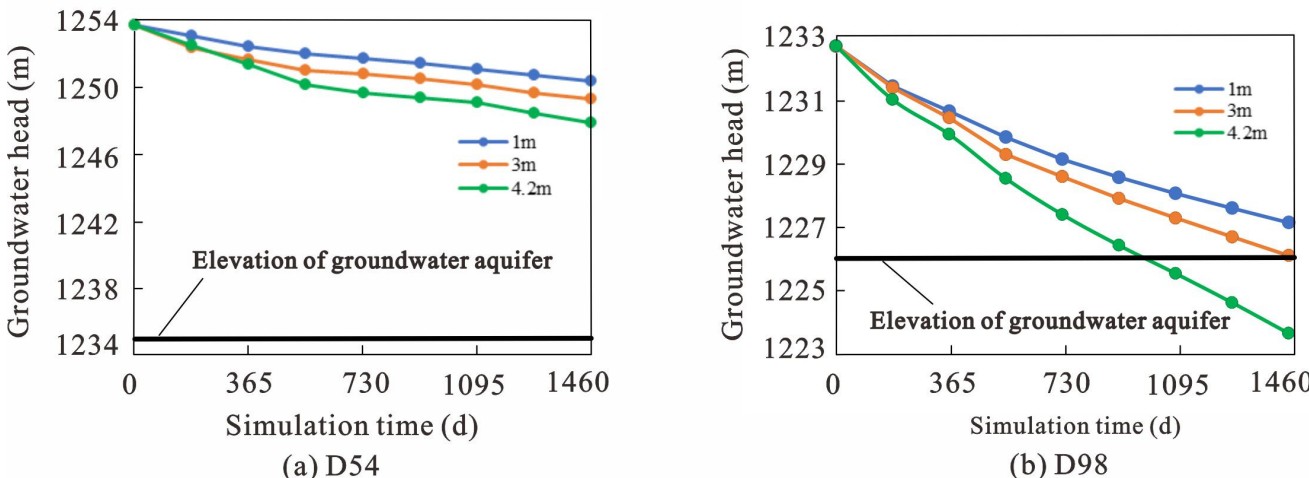

**Figure 9.** Variation in groundwater level in the upper aquifer system under different thicknesses of the coal seam: (**a**) D54, (**b**) D98.

As can be seen in Figure 10, the change in water level in the lower aquifer system is not obvious. The results are as follows. (1) The stratum is a 2–2 coal seam, and the goaf is formed in this area after coal seam mining. The lithology of the stratum did not change in this simulation, and the relevant hydrogeological parameters did not change under different conditions. (2) The variables in the simulation were the height of the caving zone and the height of the water-flowing fractured zone. The height of the water-level in the underground reservoir is the actual water storage height. The water-level height under different conditions is consistent, so the area is not affected.

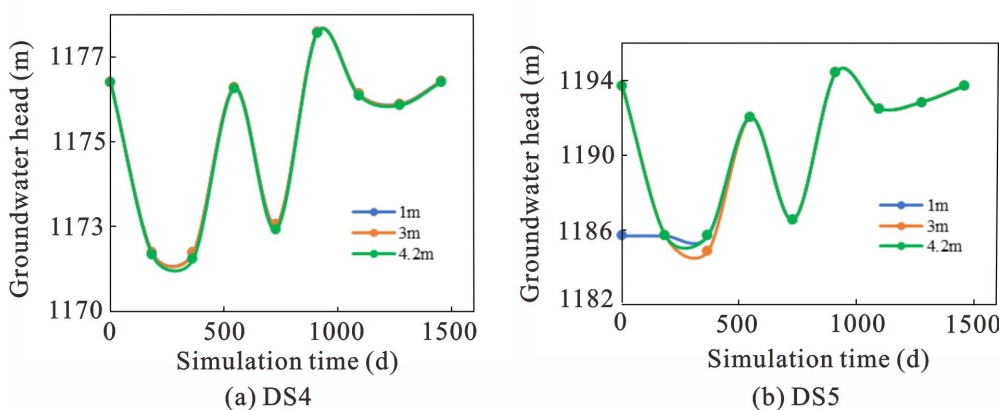

**Figure 10.** Variation in groundwater level in the lower aquifer system under different thicknesses of coal seam: (**a**) DS4, (**b**) DS5. Note: As the water-level change is not obvious, these graphs are zoomed in.

### 4.2. Impact of Overlying Lithology

The overlaying lithology of an underground reservoir determines the channel conditions of the water supply [9,19,25]. According to the overlaying lithology types of other

mined coal seams in the study region, we simulated the changes in the groundwater system in the study area under three overlying lithology conditions: hard, medium–hard, and very weak. The simulation results are shown in Figure 11.

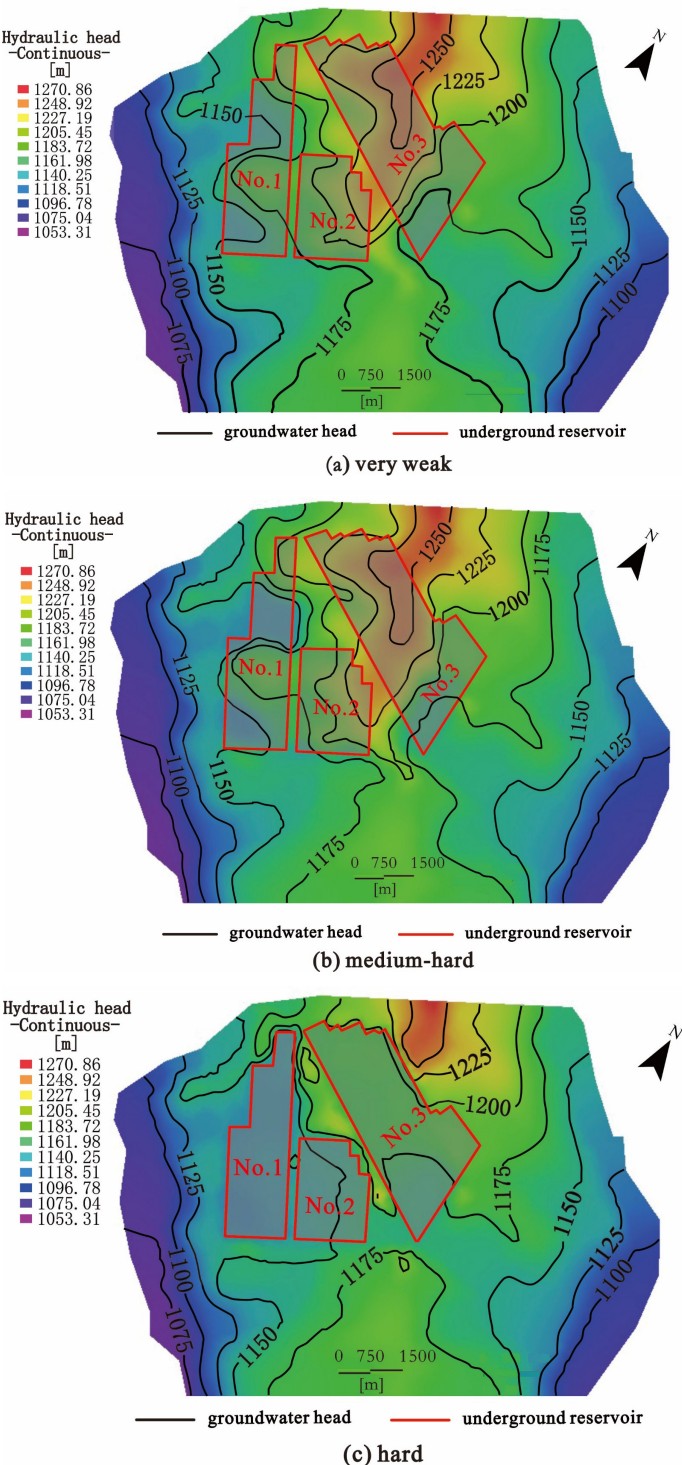

**Figure 11.** Groundwater contour map of the upper aquifer system under different lithologies of overlying strata: (**a**) very weak, (**b**) medium-hard, (**c**) hard.

As can be seen in Figures 11–13, when the overlying strata have hard lithologies, this has the greatest influence on the upper aquifer system, and the lower aquifer system has

little influence on the groundwater flow system in the reservoir area due to the impacts of reservoir water storage factors.

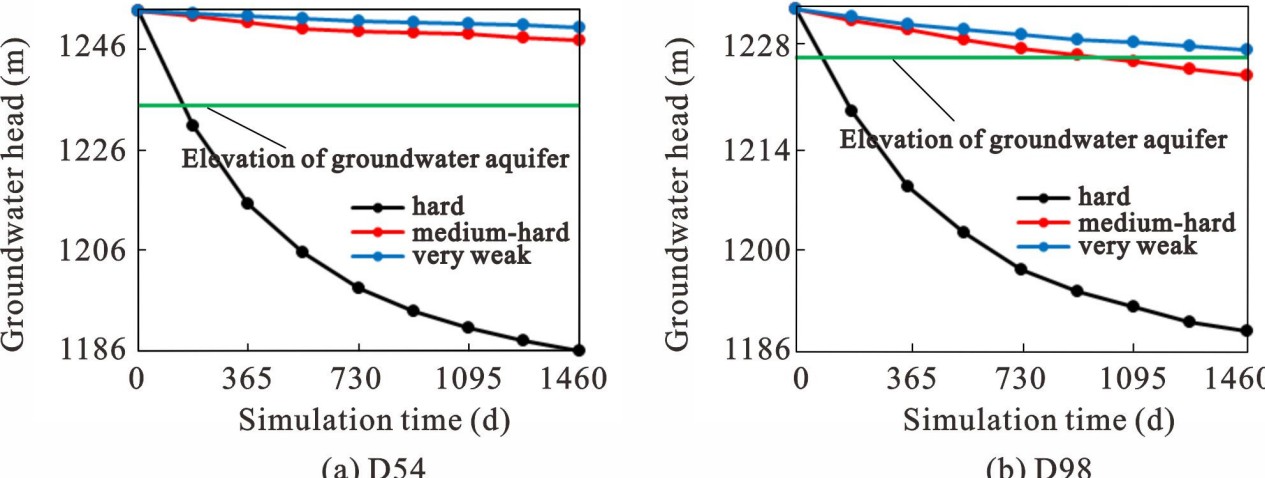

**Figure 12.** Variation in groundwater level in the upper aquifer system under different lithologies of overlying strata: (**a**) D54, (**b**) D98.

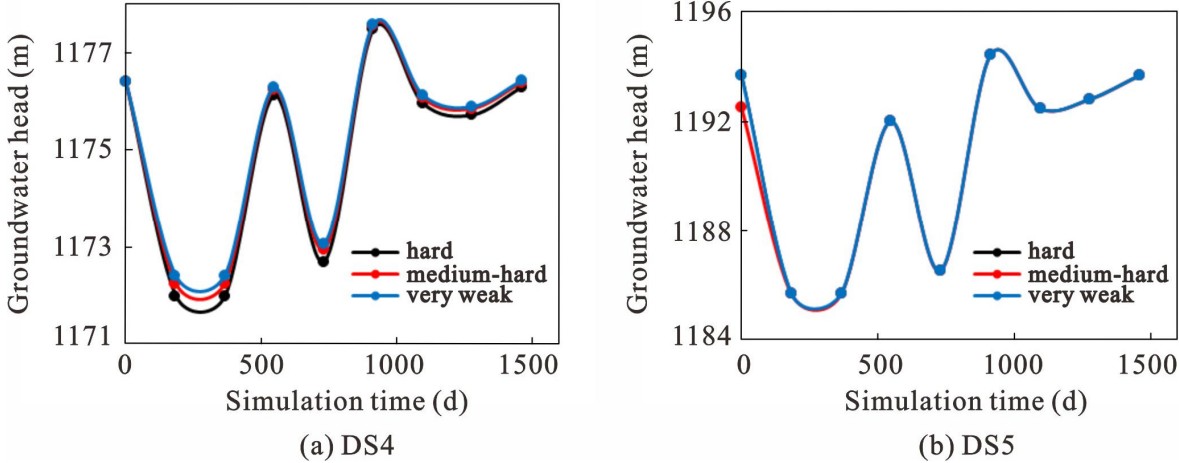

**Figure 13.** Variation in groundwater level in the lower aquifer system under different lithologies of overlaying strata: (**a**) DS4, (**b**) DS5. Note: As the water level change is not obvious, these graphs are zoomed in.

Based on the water-level changes in D54 and D98 under different lithology conditions (Figure 12), the following can be concluded. (1) When the overlying lithology is hard, the upper aquifer system's water level decreases dramatically. On day 1460, the drawdown values of boreholes D54 and D98 are 67.66 and 43.87 m, respectively, which are less than the restricting lower bed elevation of the aquifer. (2) In the first month, the drawdown values are 4.62 and 2.67 m, respectively. (3) The decline in the groundwater level in the upper aquifer system reduces progressively as the lithology of the overlaying rock transitions from hard to very weak. The aquifer's drainage gradually diminishes until it can no longer be drained.

### 4.3. Impact of the Water-Storage Range

The water-storage range of underground reservoirs is one of the parameters that determine the reservoirs' water storage capacity [25–27]. The tunneling depth of the workface was simulated as 2000 m or 6000 m, that is, the changes in the groundwater flow

system were simulated in the mining area for the water-storage ranges of 2000 m × 400 m and 6000 m × 400 m—and the simulation results are shown in Figures 14 and 15.

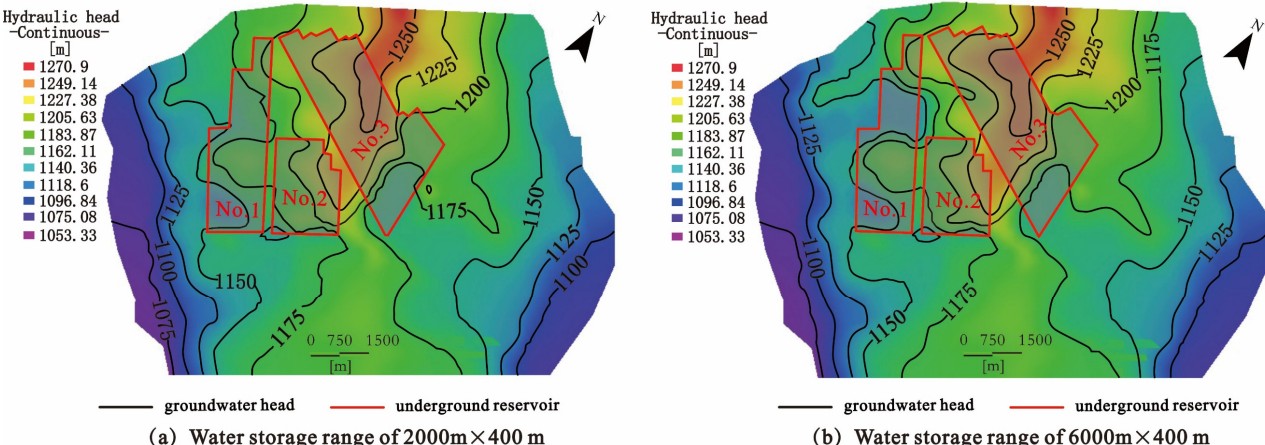

(a) Water storage range of 2000m×400 m

(b) Water storage range of 6000m×400 m

**Figure 14.** Groundwater contour map of the upper aquifer system under different water-storage ranges: (**a**) 2000 × 400 m, (**b**) 6000 × 400 m.

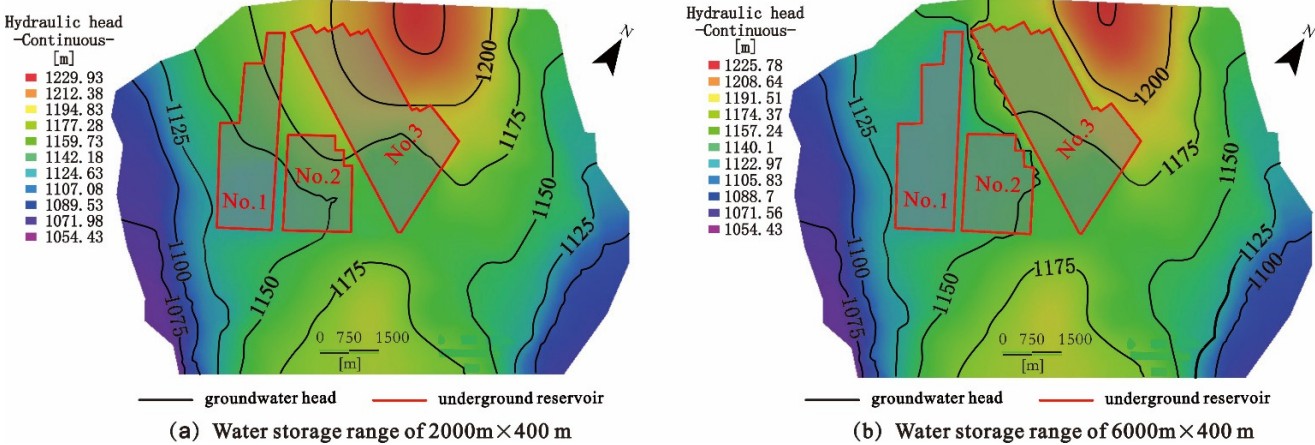

(a) Water storage range of 2000m×400 m

(b) Water storage range of 6000m×400 m

**Figure 15.** Groundwater contour map of the lower aquifer system under different water-storage ranges: (**a**) 2000 × 400 m, (**b**) 6000 × 400 m.

As seen in Figures 14 and 15, the groundwater system in the mining area is impacted more when the water-storage range of the underground reservoir increases. The lower aquifer system is impacted more than the upper aquifer system, and the water-storage range of the underground reservoir affects the runoff and discharge range of the groundwater system in the mining area and modifies the initial conditions of the supplementary runoff. For instance, when the driving depth increases in the western portion of the coal seam (Figure 15), the original runoff area eventually transforms into the discharge area.

According to the variation in the water level in borehole DS3 (Figure 16), when the water-storage range of the workface was 2000 m × 400 m, the groundwater level of the lower aquifer system rose with time, and the biggest increase, 14.01 m, occurred during the first month. When the water-storage range was 6000 m × 400 m, the water level remained consistent throughout the duration of the simulation, mostly due to the proximity of borehole DS3 to the first reservoir. When the water-storage range reached 6000 m × 400 m, the borehole water level corresponded to the water level of the underground reservoir.

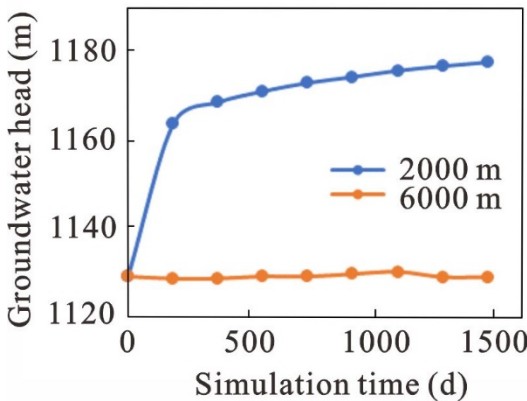

**Figure 16.** Variation in water level in borehole D54 for different water-storage ranges.

*4.4. Repercussions of Storage Water Level*

Due to the recharge, runoff, and discharge relationships among underground reservoirs in the study area, to reduce the mutual influence among underground reservoirs in the simulation process, we analyzed the water levels of underground reservoirs using only Underground Reservoir 1 as the research object. We chose the water storage height that matched the underground reservoir's characteristic storage capacity for study and analysis. In this paper, 5 m and 10 m of groundwater reservoir storage level were selected for the study. According to Gao et al. [17], the average water storage coefficient is 0.25 when the water level of the underground reservoir is 5 m, and it is 0.14 when the water level is 10 m. The simulation results are shown in Figures 17 and 18.

As seen in Figures 17 and 18, when the water level of the underground reservoir changes, the upper aquifer system is largely unaffected and the water level in the goaf does not rise significantly. As the water level of the underground reservoir increases, the overall groundwater level of the lower aquifer system rises. This implies that the underground reservoir's water storage has a "water retention" effect on the groundwater system, enabling the storage of groundwater and a reduction in water resource loss. The variation in the water level of the underground reservoir has a greater effect on the groundwater of the aquifer where the reservoir is located than it does on the upper aquifer system, particularly on the goaf.

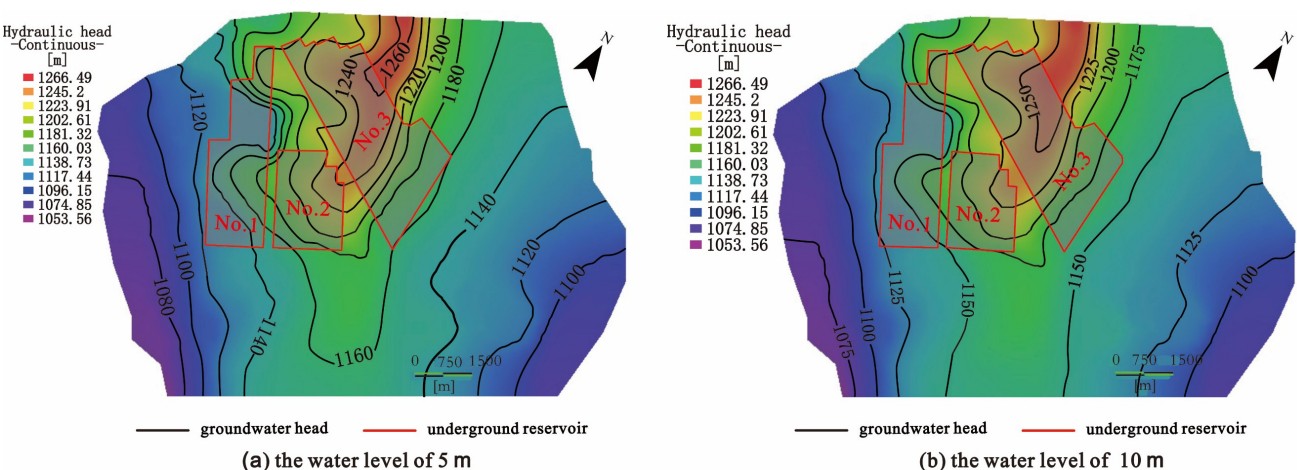

**Figure 17.** Groundwater contour map of the upper aquifer system for different impoundment heights of the underground reservoir: (**a**) 5 m, (**b**) 10 m.

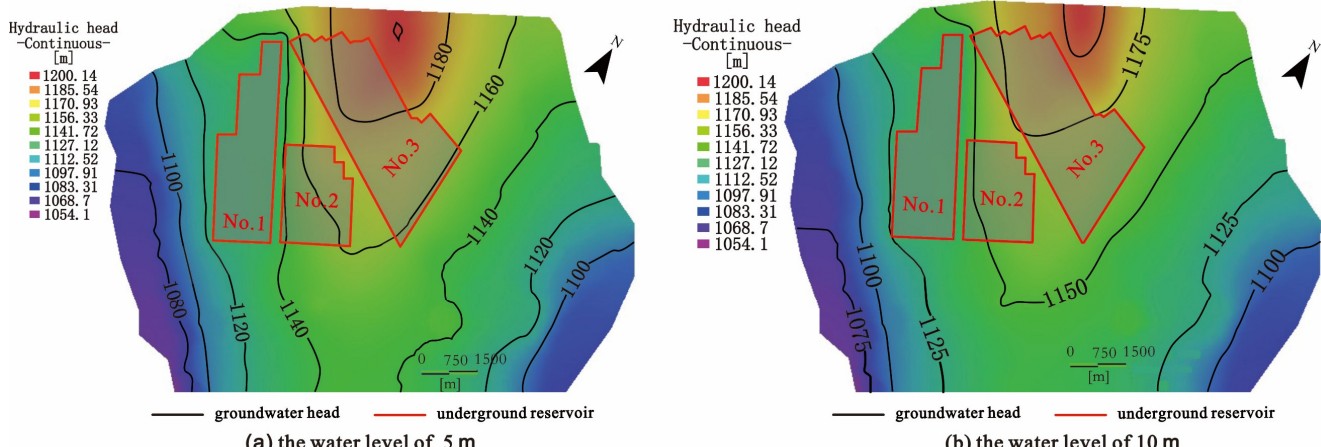

**Figure 18.** Groundwater contour map of the lower aquifer system for different impoundment heights of the underground reservoir: (**a**) 5 m, (**b**) 10 m.

*4.5. Sensitivity Analysis of Impacting Factors*

Analysis of the sensitivity of impacting factors can enable identification of the degree of impact of each factor on the building of underground reservoirs and provide a foundation for the future construction of underground reservoirs and the protection of regional water resources. We employed the technique of local sensitivity analysis. When computing the sensitivity coefficient of a specific parameter, other parameters are expected to remain unaltered, and each parameter is considered independent of the others by default [28]; there are changes to a specific parameter while other parameters are unchanged. The sensitivity coefficient of the parameter to the dependent variable is calculated by dividing the change in the value of the dependent variable by the change in the quantity of the parameter [29,30]. The equation used for the sensitivity analysis is as follows:

$$X_{i,k} = \frac{\partial \hat{y}}{\partial a_k} \approx \frac{\hat{y}_i(a_k + \Delta a_k) - \hat{y}_i(a_k)}{\Delta a_k / a_k} \tag{1}$$

$\hat{y}_i$ is the dependent variable, $X_{i,k}$ is the sensitivity coefficient of the dependent variable based on the *K*th parameter at the *i*th monitoring point, $a_k$ is the value of the parameter in the basic case, and $\Delta a_k$ is the amount of disturbance applied to this parameter.

The sensitivity coefficients calculated for each impacting factor are presented in Table 4. Table 4 illustrates that the mining thickness of the coal seam and the lithology of the overlying reservoir at the location of the underground reservoir have a major impact on the upper aquifer system. Both the storage capacity and water level of an underground reservoir have a substantial impact on the lower aquifer system.

**Table 4.** Absolute sensitivity coefficient values for each influencing factor of the observation well.

| Affecting Factors | Observation Well | | | | | | |
|---|---|---|---|---|---|---|---|
| | **D54** | **D98** | **D82** | **D81** | **DS2** | **DS3** | **DS5** |
| coal seam mining thickness | 4.795 | 8.645 | 0.035 | 0.070 | - | 0.945 | - |
| overlying strata | 3.040 | 4.280 | 0.002 | 0.003 | - | 0.888 | - |
| water-storage range | 3.960 | 1.830 | 0.003 | 0.060 | - | 7.520 | - |
| storage water level | 1.050 | 0.050 | 0.050 | 0.100 | 5.000 | 4.800 | 1.060 |

## 5. Conclusions

In this study, a comprehensive hydrogeological numerical model for a mining area and underground reservoir was established for the first time. Factors such as the mining thickness of the coal seam at the location of the underground reservoir, the water-storage

range of the reservoir, the overlying lithology of the reservoir, and the water storage level were used to analyze the impact of the construction of the underground reservoir on the flow field in the mining area, and the sensitivity coefficients of each influencing factor were calculated. The following conclusions were reached.

(1) The mining thickness of the coal seam and the overlying lithology of the reservoir are the main factors affecting the flow system of the upper aquifer system. Under the condition of goaf water storage, the development height of the "upper three zones" increases with the increase in coal seam mining thickness, and the water level of the upper aquifer system decreases. In contrast, the groundwater flow system of the lower aquifer system changes slightly. By comparing the changes in the flow system in the mining area under the conditions of hard, medium–hard, and very weak overlying lithology, it was found that the hard overlying lithology has the most significant impact on the flow of the upper aquifer system.

(2) Water-storage range and reservoir height are the main factors affecting the groundwater system of the lower aquifer system. As the water-storage range of the underground reservoir increases, the influence of the construction of the underground reservoir on the groundwater system of the mining area also increases. The water-storage range of the underground reservoir changes the initial recharge, runoff, and discharge conditions, affecting the runoff and discharge range of the groundwater system in the study area. When the water level of the underground reservoir gradually increases from 5 to 10 m, it significantly impacts the lower aquifer system, especially the groundwater flow in the goaf area.

(3) As the underground water level increases, the water level of the lower aquifer system increases significantly, and the water level of the upper aquifer system also increases. This proves that the underground reservoir storage has a positive effect on water retention in the groundwater system in the mining area, achieving the desired results of storing groundwater and reducing water loss, which has a positive effect on the mining area's environment.

(4) For the fine characterization of the special underground hydraulic engineering of underground reservoirs, the biggest problems encountered in the study were the prominent non-homogeneity of its spatial structure, the obvious three-dimensional flow characteristics, and the difficulty of parameter quantification. Although underground reservoirs have been characterized to a certain extent in previous studies, there is still some room for improvement.

(5) The underground reservoir in the mining area is not a single reservoir structure, but a group of reservoirs composed of several underground reservoirs, and there is also a certain hydraulic connection between each reservoir, which creates a complex groundwater system. The evolution of the water flow system in the mine area, especially when the reservoirs are not contemporaneously influenced by each other, is also worthy of in-depth exploration and study.

(6) In future research, the water quality model can be superimposed on the groundwater flow model to analyze the changes in the water quality of the upper pore water and the pit reinjection water after flowing through the groundwater reservoir under different conditions in the groundwater reservoir. This is also a problem to be addressed in respect of the construction and operation of groundwater reservoirs.

**Author Contributions:** Conceptualization, L.W.; methodology, L.W. and Y.C.; software, F.X.; validation, Y.C. and Y.Z.; formal analysis, L.W. and Z.X.; investigation, Y.Z.; resources, L.W. and Y.C.; data curation, Z.X.; writing—original draft preparation, Y.C. and L.W.; writing—review and editing, Y.Z.; visualization, Y.C. and Y.Z.; supervision, L.W.; project administration, L.W.; funding acquisition, L.W. All authors have read and agreed to the published version of the manuscript.

**Funding:** This research was funded by the State Key Laboratory of Water Resource Protection and Utilization in Coal Mining Open Fund Project, grant number: SHJT-17-42.9, and the Natural Science Foundation of Hubei Province of China, grant number: 2020CFB750.

**Institutional Review Board Statement:** Not applicable.

**Informed Consent Statement:** Not applicable.

**Data Availability Statement:** Not applicable.

**Conflicts of Interest:** The authors declare no conflict of interest.

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
