# Peer review of "Impacts of Underground Reservoir Site Selection and Water Storage on the Groundwater Flow System in a Mining Area—A Case Study of Daliuta Mine"

_water, doi:10.3390/w14203282_

Round 1

Reviewer 1 Report

Review of “Study on the impact of underground reservoir site selection and water 2 storage on the groundwater flow system in mining area-- Taking Da-3 liuta Coal in Shendong Mining Area as an example”

Overall, the paper suffers greatly from typos and grammar which interferes with the reviewer focusing on the science presented.  Had the paper gotten a better editing it would greatly improve its readability.  The paper also suffers from poor figure production, while the model effort is interesting the lack of communication through good maps and figure also hinders the paper.  I recommend a major rewrite and recasting of the figures before submission.

Where did the hydraulic conductivity values come from?  Earlier studies, a guess or estimation?  No clear discussion provided the reader, yet the model depends upon these values.  What is the vertical versus horizontal isotropy? 

This paper needs a major revision overall with an expanded hydrogeologic unit description along with the lithology and hydraulic conductivity and the source of this data.  

Here are some specific comments:

In the first line of the abstract the term water resource is mentioned without definition of what is meant by it.

Line 17  the word takes should be replaced by “uses”

Line 18  Use the word “determine” instead of discuss.

Line 20  Use the term “coal seam mining thicknesses”

The references are not in alphabetical order.

Line 42  Use “altering” instead of alters.

Line 42-46 This is a run on sentence please break it up.

Line 63, 67 Add the word “the” before the noun.

Line 75 Where is the location figure?  You are describing the study area and the reader has no figure to follow what is being discussed.

Line 76-77 The discussion on terrain makes no sense since you do not quantify the amount of relief or give a range for elevation.

Line 79 Sentence is missing the word “is”.

Line 84-97 The discussion of geology should be defined as such not strata.  No real descriptions are provided for the geology nor is there any attempt to describe the units and their materials here.  The paper suffers from a lack of geologic information. The table 1 provides some information but what is “expand the bottom”?  This entire section needs to be rewritten. Table 1 stratum number should be model layer number.

No discussion in the text on the hydrogeologic units and their geology.

Table 1 also has no units provided to the reader regarding hydraulic conductivity.  The paper never seems to discuss hydraulic conductivity of the hydrogeologic units.  

No discussion on the upper aquifer system and the lower aquifer system as defined in table 1.

Line 101 What is meant by the term “upper three zones” it is not described and explained which is very confusing to the reader.

Line 104 How does a water-resistant coal seam become an aquifer? This is very confusing and contributes to the poor readability of the paper.

Line 111 add the word “a”.

Line 117 The term “F6 fault is suddenly introduced with no explanation and no reference to any figure shown.  This is another example of the poor handle on the geology of the area.

Line 121 Figure 1 has no scale, no legend describing any numbered dots, no latitude or longitude, no larger inset map to show where the study is in reference to the entire country or world. No plotting of where the fault F6 is located.

Line 123 Add the word “the” before vertical direction and stratigraphic.

Figure 2 is not readable regarding the 14 layers of the hydrogeologic units and their relation to model layers.

Table 2 is not clear since units are not provided and there are typos.

Figure 3 makes no sense since no scale, no legend, no latitude or longitude, no reference of any mines or landmarks, and no reference to what the numbers mean.

Table 4 needs definition of what Kzz is referring to, and significant figures are not used with the values presented.

Line 139 Fracture rate is introduced with no discussion on what it means or what units it is.  There is no discussion of bedrock fractures or faults in the geology section of the paper. Again, the lack of a clear handle on the geology confuses the reader and indicates the model is not tied directly to the geology.

Line 143 Sentence has poor grammar.

Line 150 What is meant by geophysical exploration here? No explanation is provided.

Line 181 Should reference figure 1.  These wells should have been introduced early in the paper not at the end.  There should have been a clearly defined hydrogeologic section in the beginning of the paper including each layer, it’s geology and materials, its hydraulic conductivity, and where the hydraulic conductivity value came from?

Line 201 Hydraulic conductivity should be a major factor in the groundwater flow system but is not mentioned.

Line 245 The term intrusive rock is introduced here.  I have no idea what the authors mean by introducing a magmatic rock term here.

Figures 4, 7, 10, 14, and 17 are all suffering from lack of significant figures in the legend, no scale, no latitude or longitude, no clear indication of where the wells are located, no clear indication of where the mines are located referenced in figure 1. 

Conclusions are confusing since the terms groundwater system and underground reservoir are used.  What happened to the terms upper and lower aquifer system introduced in table 1.  This is very confusing and again seems to indicate a focus on building a model but no clear connection to the geologic framework and the two major aquifer systems.

Reviewer 2 Report

Reviewer Comments

Paper title: Study on the impact of underground reservoir site selection and water storage on the groundwater flow system in mining area-- Taking Daliuta Coal in Shendong Mining Area as an example

The present manuscript describes the research result of using the FEFLOW software to construct a groundwater flow numerical model in the mining area including underground reservoir. Also discussed the impact of different site selection and water storage factors on groundwater system. The model was used to analyze the impact of mining thickness of coal seam, overlying lithology, water storage range, and water level of the underground reservoir on the groundwater flow system in the mining area.

A manuscript has a practical application and also provides important theoretical for the next studies.

The paper can be accepted for publication after providing the corrections mentioned below.

Point 1. The abstract section sounds unclear. The abstract should follow the MDPI style of structured abstracts:

- Background (place the question addressed in a broad context and highlight the purpose of the study);

- Methods (describe briefly the main methods);

- Results (summarize the article's main findings);

- Conclusion (indicate the main conclusions or interpretations).

Point 2. Keywords need to be modified. Please use words not combinations of words or phrases.

Point 3. In the Introduction section, an enhanced literature review is required. For this study, the authors have used only 25 literature sources (all references come from the China). It seems insufficient for such type of research. What about foreign experience in this direction?

Point 4. It will be great if the authors show some description in context – Why it is important to conduct this study? Can the expected result be used or implemented within other coal basins? If yes, then how? What limitations?

Point 5. The aim and the tasks must be highlighted at the end of the Introduction section.

Point 6. You can use Case Study instead of Overview of Study Area. It seems better.

Point 7. Can you show a map of China where study area is located?

Point 8. What is measures of Hydraulic conductivity in the table 1?

Point 9. Table 2 precipitatioN. N is missed.

Point 10. Why only thickness of 1, 3 and 4.2m are studied?

Point 11. The novelty of the paper must be highlighted in the conclusions section.

Point 12. Please provide a short description of further research.

Point 13. There are papers that I have reviewed in the past years. Please consider the suggested research in your paper when enhancing the literature review*. I believe they are worth considering in your paper.

Rudakov, D., & Westermann, S. (2021). Analytical modeling of mine water rebound: Three case studies in closed hard-coal mines in Germany. Mining of Mineral Deposits, 15(3), 22-30. https://doi.org/10.33271/mining15.03.022

Sadovenko, I., Ulytsky, O., Zahrytsenko, А., & Boiko, K. (2020). Risk assessment of radionuclide contamination spreading while flooding coal mined-out rocks. Mining of Mineral Deposits, 14(4), 130-136. https://doi.org/10.33271/mining14.04.130

Bazaluk, O., Sadovenko, I., Zahrytsenko, A., Saik, P., Lozynskyi, V., & Dychkovskyi, R. (2021). Forecasting Underground Water Dynamics within the Technogenic Environment of a Mine Field: Case Study. Sustainability, 13(13), 7161. https://doi.org/10.3390/su13137161

Rudakov, D., & Inkin, O. (2022). Evaluation of heat supply with maintaining a safe mine water level during operation of open geothermal systems in post-coalmining areas. Mining of Mineral Deposits, 16(1), 24-31. https://doi.org/10.33271/mining16.01.024

Point 14. In general, I must admit that a very good study was performed, and I will recommend your paper for publication after careful revision.

Round 2

Reviewer 1 Report

The authors have done a good job addressing my concerns and making improvements to their paper. The paper appears to be in good shape.

Reviewer 2 Report

Dear authors,

I am more than satisfied with the corrections provided by you.

This study is an important contribution to sustainable mining.

Congratulations to the authors.